# Revisiting Machine Translation for Cross-lingual Classification

**Mikel Artetxe**[1,2*]   **Vedanuj Goswami**[3]   **Shruti Bhosale**[3]
**Angela Fan**[3]   **Luke Zettlemoyer**[3]
[1]Reka AI   [2]HiTZ Center, University of the Basque Country (UPV/EHU)   [3]Meta AI
mikel@reka.ai   {vedanuj,shru,angelafan,lsz}@meta.com

## Abstract

Machine Translation (MT) has been widely used for cross-lingual classification, either by translating the test set into English and running inference with a monolingual model (*translate-test*), or translating the training set into the target languages and finetuning a multilingual model (*translate-train*). However, most research in the area focuses on the multilingual models rather than the MT component. We show that, by using a stronger MT system and mitigating the mismatch between training on original text and running inference on machine translated text, *translate-test* can do substantially better than previously assumed. The optimal approach, however, is highly task dependent, as we identify various sources of cross-lingual transfer gap that affect different tasks and approaches differently. Our work calls into question the dominance of multilingual models for cross-lingual classification, and prompts to pay more attention to MT-based baselines.

## 1 Introduction

Recent work in cross-lingual learning has pivoted around multilingual models, which are typically pretrained on unlabeled corpora in multiple languages using some form of language modeling objective (Doddapaneni et al., 2021). When fine-tuned on downstream data in a single language—typically English—these models are able to generalize to the rest of the languages thanks to the aligned representations learned at pretraining time, an approach known as ***zero-shot*** transfer. The so called ***translate-train*** is an extension of this method that augments the downstream training data by translating it to all target languages through MT. A third approach, ***translate-test***, uses MT to translate the test data into English, and runs inference using an English-only model.

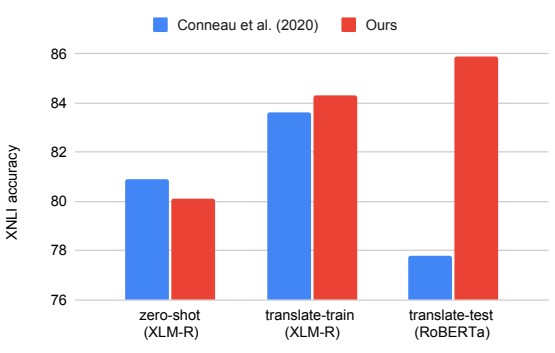

Figure 1: **XNLI accuracy.** We show that *translate-test* can do substantially better than previously reported.

The conventional wisdom is that *translate-train* tends to bring modest improvements over *zero-shot*, while *translate-test* has been largely overlooked and is often not even considered as a baseline (Hu et al., 2020; Ruder et al., 2021). In this context, most recent research in multilingual NLP has focused on pretraining stronger multilingual models (Conneau et al., 2020; Xue et al., 2021; Chi et al., 2022), and/or designing better fine-tuning techniques to elicit cross-lingual transfer (Liu et al., 2021; Yu and Joty, 2021; Zheng et al., 2021; Fang et al., 2021). However, little attention has been paid to the MT component despite its central role in both *translate-train* and *translate-test*. Most authors use the official translations that some multilingual benchmarks come with, and it is unclear the extent to which better results could be obtained by using stronger MT systems or developing better integration techniques.

In this work, we revisit the use of MT for cross-lingual learning through extensive experiments on 6 classification benchmarks. We find evidence that *translate-test* is more sensitive than *translate-train* to the quality of the MT engine, and show that better results can be obtained by mitigating the mismatch between training on original downstream data and running inference on machine translated

---

*Work done at Meta AI

data. As exemplified by Figure 1, we demonstrate that, with enough care, *translate-test* can work substantially better than previously assumed (e.g., outperforming both *zero-shot* and *translate-train* on XNLI for the first time), calling into question the dominance of multilingual pretraining in the area.

However, these trends are not universal, as we find that the optimal approach is highly task dependent. We introduce a new methodology to quantify the underlying sources of cross-lingual transfer gap that cause this discrepancy across tasks. We find that *translate-test* excels on complex tasks requiring commonsense or real world knowledge, as it benefits from the use of a stronger English-only model. In contrast, *translate-train* performs best at shallower tasks like sentiment analysis, for which the noise introduced by MT outweighs the benefit of using a better pretrained model.

## 2 Experimental setup

**Pre-trained models.** We use RoBERTa large (Liu et al., 2019) and XLM-R large (Conneau et al., 2020) as our primary English-only and multilingual models, respectively. XLM-R can be considered the multilingual version of RoBERTa,[1] as they are both encoder-only masked language models trained with the same codebase. So as to understand the benefits from using a stronger English-only model, we also experiment with DeBERTaV3 large (He et al., 2021). All the 3 models have 304M backbone parameters, although they differ in the size of their vocabulary.

**Machine translation.** We use the 3.3B NLLB model (NLLB Team et al., 2022) as our MT engine. We explore two decoding strategies as indicated in each experiment: beam search with a beam size of 4, and nucleus sampling (Holtzman et al., 2020) with top-$p$ = 0.8. Unless otherwise indicated, we translate at the sentence level using the xx_sent_ud_sm model in spaCy for sentence segmentation.[2] For variants translating at the document level, we concatenate all fields (e.g., the premise and the hypothesis in XNLI) using a special word <sep> to separate them. For experiments involving MT fine-tuning, we use a learning rate of 5e-05 and a batch size of 32k tokens with dropout disabled, and use the final checkpoint after 25k steps.

**Evaluation.** We use the following 6 datasets for evaluation: **XNLI** (Conneau et al., 2018), a Natural Language Inference (NLI) dataset covering 15 languages; **PAWS-X** (Yang et al., 2019), an adversarial paraphrase identification dataset covering 7 languages; **MARC** (Keung et al., 2020), a sentiment analysis dataset in 6 languages where one needs to predict the star rating of an Amazon review; **XCOPA** (Ponti et al., 2020), a causal commonsense reasoning dataset covering 12 languages; **XStoryCloze** (Lin et al., 2021), a commonsense reasoning dataset in 11 languages where one needs to predict the correct ending to a four-sentence story; and **EXAMS** (Hardalov et al., 2020), a dataset of high school multiple choice exam questions in 16 languages. For a fair comparison between different approaches, we exclude Quechua and Haitian Creole from XCOPA, as the former is not supported by NLLB and the latter is not supported by XLM-R. In all cases, we do 5 finetuning runs with different random seeds, and report accuracy numbers in the test set averaged across all languages and runs. Following common practice, we use MultiNLI (Williams et al., 2018) as our training data for XNLI, and PAWS (Zhang et al., 2019) as our training data for PAWS-X. For MARC, we use the English portion of the training set. XCOPA, XStoryCloze and EXAMS do not come with a sizable training set in English, so we train on the combination of the following multiple choice datasets: Social IQa (Sap et al., 2019), SWAG (Zellers et al., 2018), COPA (Roemmele et al., 2011), OpenBookQA (Mihaylov et al., 2018), ARC (Clark et al., 2018) and PIQA (Bisk et al., 2020).

**Fine-tuning.** We use HuggingFace's Transformers library (Wolf et al., 2019) for all of our experiments. So as to handle a variable number of candidates in multiple choice tasks (XCOPA, XStoryCloze and EXAMS), we feed each input-candidate pair independently into the model, take its final representation from the first token, down-project into a scalar score through a linear projection, and apply the softmax function over the scores of all candidates. For the remaining tasks, we simply learn a regular classification head. In all cases, we use a batch size of 64 and truncate examples longer than 256 tokens. We train with a learning rate of 6e-6 with linear decay and 50 warmup steps, and use the final checkpoint without any model selection. When using the original English training set, we finetune for two epochs. For experiments

---

[1] In fact, XLM-R stands for XLM-RoBERTa.
[2] https://spacy.io/models/xx#xx_sent_ud_sm

involving some form of data augmentation, where each training example is (back-)translated into multiple instances, we finetune for a single epoch.

## 3 Main results

We next present our main results revisiting MT for cross-lingual classification, both for *translate-test* (§3.1) and *translate-train* (§3.2). Finally, we put everything together and reconsider the state-of-the-art in the light of our findings (§3.3).

### 3.1 Revisiting translate-test

While *translate-test* might be regarded as a simple baseline, we argue that there are two critical aspects of it that have been overlooked in prior work. First, little attention has been paid to the MT engine itself: most existing works use the official translations from each dataset without any consideration about their quality, and the potential for improvement from using stronger MT systems is largely unknown. Second, *translate-test* uses original (human generated) English data to fine-tune the model, but the actual input that is fed into the model at test time is produced by MT. Prior work has shown that original and machine translated data have different properties, and this mismatch is detrimental for performance (Artetxe et al., 2020), but there is barely any work addressing this issue. In what follows, we report results on our 6 evaluation tasks using a strong MT system, and explore two different approaches to mitigate the train/test mismatch issue: adapting MT to produce translations that are more similar to the original training data (§3.1.1), and adapting the training data to make it more similar to the translations from MT (§3.1.2).

#### 3.1.1 MT adaptation

Table 1 reports *translate-test* results with RoBERTa using translations from 4 different MT systems: the official ones from each dataset (if any), the vanilla NLLB 3.3B model, and two fine-tuned variants of it. In the first variant (+*dom adapt*), we segment the downstream training data into sentences, backtranslate them into all target languages using beam search, and fine-tune NLLB on it as described in §2. The second variant (+*doc level*) is similar, except that we concatenate the back-translated sentences first, and all the input fields (e.g. the premise and the hypothesis) afterwards, separated by <sep>, and fine-tune NLLB on the resulting synthetic parallel data. For this last variant we also translate at

| MT engine | xnli | pwsx | marc | xcop | xsto | exm |
|---|---|---|---|---|---|---|
| Official | 76.8 | – | – | 75.9 | – | – |
| NLLB | 79.9 | 87.3 | 57.6 | 72.9 | 89.3 | 36.3 |
| + dom adapt | 80.8 | 86.9 | 58.2 | 72.9 | 88.4 | **36.5** |
| + doc level | **83.8** | **87.8** | **58.3** | **76.3** | **91.3** | 36.3 |

Table 1: Translate-test results with different MT engines. All variants use a RoBERTa model finetuned on the original English data.

the document level at test time, whereas the rest of the systems translate at the sentence level.[3]

We find that document-level fine-tuned NLLB obtains the best results across the board, obtaining consistent improvements over vanilla NLLB in all benchmarks except EXAMS. We remark that the former does not have any unfair advantage in terms of the data it sees, as it leverages the exact same downstream data that RoBERTa is finetuned on. Sentence-level fine-tuned NLLB is only able to outperform vanilla NLLB on XNLI, MARC and EXAMS, and does worse on PAWS-X and XStoryCloze. This shows that a large part of the improvement from fine-tuning NLLB can be attributed to learning to jointly translate all sentences and fields from each example.

Finally, we also find a high variance in the quality of the official translations in the two datasets that include them. More concretely, the official translations are 3.0 points better than vanilla NLLB on XCOPA, and 3.1 points worse on XNLI. Despite overlooked to date, we argue that this factor has likely played an important role in some of the results from prior work. For instance, Ponti et al. (2020) obtain their best results on XCOPA using *translate-test* with RoBERTa, whereas Conneau et al. (2020) find that this approach obtains the worst results on XNLI. These seemingly contradictory findings can partly be explained by the higher quality of the official XCOPA translations, and would have been less divergent if the same MT engine was used for both datasets.

#### 3.1.2 Training data adaptation

We experiment with a form of data augmentation where we translate each training example into another language and then back into English.[4] The

---

[3]We also tried translating at the document level with the rest of the systems, but this worked poorly in our preliminary experiments, as the model would often only translate the first sentence and ignore the rest.

[4]In the forward (out-of-English) direction, we use beam search for multiplechoice tasks (XCOPA, XStoryCloze, EX-

| Train data | xnli | pwsx | marc | xcop | xsto | exm |
|---|---|---|---|---|---|---|
| Original | 79.9 | 87.3 | 57.6 | 72.9 | 89.3 | 36.3 |
| Roundtrip MT | 85.2 | **89.9** | 58.8 | 74.3 | **91.2** | 36.2 |
| + MT adapt (doc) | **85.9** | 89.3 | **59.1** | **75.7** | **91.2** | **36.4** |

Table 2: Translate-test results different training data. All methods use vanilla NLLB for translation.

| MT engine | xnli | pwsx | marc | xcop | xsto | exm |
|---|---|---|---|---|---|---|
| None (zero-shot) | 80.1 | 87.1 | 60.6 | **69.1** | 84.6 | **36.0** |
| Official | 83.3 | **90.7** | – | – | – | – |
| NLLB (beam) | 83.5 | 90.4 | 60.5 | 68.5 | 86.4 | **36.0** |
| NLLB (samp) | **84.3** | **90.7** | **60.8** | 67.6 | **86.7** | 35.2 |

Table 3: Translate-train results using XLM-R

resulting data is aimed to be more similar to the input that the model will be exposed to at test time, as they are both produced by the same MT model. For each task, we use all target languages it covers as the pivot, and further combine the back-translated examples with the original data in English.[5] We compare this approach to the baseline (training on the original English data), and report our results in Table 2. So as to understand how complementary training data adaptation and MT adaptation are, we also include a system combining the two, where we use the document-level fine-tuned model from §3.1.1 to translate the test examples.

We find that roundtrip MT outperforms the baseline by a considerable margin in all tasks but EXAMS. While prior work has shown that multilingual benchmarks created through (professional) translation have artifacts that can be exploited through similar techniques (Artetxe et al., 2020), it is worth noting that we also get improvements on MARC, which was not generated through translation. Finally, we observe that combining this approach with MT adaptation is beneficial in most cases, but the improvements are small and inconsistent across tasks. This suggests that the two techniques are little complementary, which is not surprising as they both try to mitigate the same underlying issue.

## 3.2 Revisiting translate-train

As seen in §3.1, the MT engine can have a big impact in *translate-test*. To get the full picture, we next explore using the same MT engines for *translate-train*, including the official translations (when available) as well as NLLB with beam search

and nucleus sampling. In all cases, we translate the downstream training data into all target languages covered by each task, and fine-tune XLM-R on this combined data (including a copy of the original data in English). We also include a *zero-shot* system as a baseline, which fine-tunes the same XLM-R model on the original English data only. Table 3 reports our results.

We find that *translate-train* obtains substantial improvements over *zero-shot* in half of the tasks (XNLI, PAWS-X and XStoryCloze), but performs at par or even worse in the other half (MARC, XCOPA and EXAMS). This suggests that *translate-train* might not be as generally helpful as found in prior work (Hu et al., 2020). In those tasks where MT does help, nucleus sampling outperforms beam search. This is in line with prior work in MT finding that sampling is superior to beam search for back-translation (Edunov et al., 2018) but, to the best of our knowledge, we are first to show that this also holds for cross-lingual classification.

Finally, we find that translation quality had a considerably larger impact for *translate-test* than it does for *translate-train*. More concretely, the XNLI accuracy gap between the official translations and vanilla NLLB was 3.1 for *translate-train*, and is only 0.2 for *translate-test* when using beam search, or 1.0 when using nucleus sampling. This suggests that the use of relatively weak MT engines in prior work might have resulted in underestimating the potential of *translate-test* relative to other approaches.

## 3.3 Reconsidering the state-of-the-art

We have so far analyzed *translate-test* and *translate-train* individually. We next put all the pieces together, and compare different pretrained models (XLM-R, RoBERTa and DeBERTa) using *zero-shot*, *translate-train* with nucleus sampling, and two variants of *translate-test*: the naive approach using vanilla NLLB, and our improved approach combining document-level MT adaptation and training data adaptation. We report our results

AMS), and nucleus sampling for the rest of the tasks. In the backward direction, we use beam search for all tasks. We made this decision based on preliminary experiments on the development set. Sampling in the forward direction produced more diverse translations, but was noisy for multiplechoice tasks, where some options are very short. The use of beam search when translating into English is consistent with the decoding method used at test time.

[5] As such, if the original dataset has $k$ examples and we have $n$ target languages, the resulting data will consist of $k * (n + 1)$ examples.

|  |  |  | xnli | pwsx | marc | xcop | xsto | exm | avg |
|---|---|---|---|---|---|---|---|---|---|
| XLM-R | zero-shot | | 80.1 | 87.1 | 60.6 | 69.1 | 84.6 | 36.0 | 69.6 |
| | translate-train | | 84.3 | **90.7** | **60.8** | 67.6 | 86.7 | 35.2 | 70.9 |
| | translate-test | vanilla | 79.3 | 86.9 | 58.0 | 68.6 | 84.8 | 34.9 | 68.8 |
| | | ours | 84.6 | 89.3 | 58.8 | 69.0 | 87.9 | 35.0 | 70.8 |
| RoBERTa | translate-test | vanilla | 79.9 | 87.3 | 57.6 | 72.9 | 89.3 | 36.3 | 70.6 |
| | | ours | 85.9 | 89.3 | 59.1 | 75.7 | 91.2 | 36.4 | 72.9 |
| DeBERTa | translate-test | vanilla | 81.0 | 87.1 | 58.2 | 77.7 | 92.1 | **46.1** | 73.7 |
| | | ours | **86.7** | 90.3 | 59.2 | **81.3** | **93.8** | 46.0 | **76.2** |

Table 4: Main results. All systems use NLLB for MT. Best model results underlined, best overall results in **bold**.

in Table 4.

We observe that our improved variant of *translate-test* consistently outperforms the vanilla approach. Interestingly, the improvements are generally bigger for stronger pretrained models: an average of 2.0 points for XLM-R, 2.3 for RoBERTa, and 2.5 for DeBERTa.

When comparing different approaches, we find that *zero-shot* obtains the worst results in average, while *translate-train* is only 0.1 points better than *translate-test* with XLM-R. However, this comparison is unfair to *translate-test*, as there is no point in using a multilingual pretrained model when one is translating everything into English at test time. When using RoBERTa (a comparable English-only model), *translate-test* outperforms the best XLM-R results by 2.0 points. Using the stronger De-BERTa model further pushes the improvements to 5.3 points.

These results evidence that *translate-test* can be considerably more competitive than suggested by prior work. For instance, the seminal work from Conneau et al. (2020) reported that RoBERTa *translate-test* lagged behind XLM-R *zero-shot* and *translate-train* on XNLI. As illustrated in Figure 1, we show that, with enough care, it can actually obtain the best results of all, outperforming the original numbers from Conneau et al. (2020) by 8.1 points. Our approach is also 3.9 points better than Ponti et al. (2021), the previous state-of-the-art for *translate-test* in this task. While most work in cross-lingual classification focuses on multilingual models, this shows that competitive or even superior results can also be obtained with English-only models.

Nevertheless, we also find a considerable variance across tasks. The best results are obtained by *translate-test* in 4 out of 6 benchmarks—in most cases by a large margin—but *translate-train*

is slightly better in the other 2. For instance, De-BERTa *translate-test* is 13.7 points better than XLM-R *translate-train* on XCOPA, but 1.6 points worse on MARC. This suggests that different tasks have different properties, which make different approaches more or less suitable.

## 4 Analyzing the variance across tasks

As we have just seen, the optimal cross-lingual learning approach is highly task dependent. In this section, we try to characterize the specific factors that explain this different behavior. To that end, we build on the concept of **cross-lingual transfer gap**, which is defined as the difference in performance between the source language that we have training data in (typically English) and the target language that we are evaluating on (Hu et al., 2020). While prior work has used this as an absolute metric to compare the cross-lingual generalization capabilities of different multilingual models, we argue that such a transfer gap can be attributed to different **sources** depending on the approach used, which we try to quantify empirically.

In what follows, we identify the specific sources of transfer gap that each approach is sensitive to (§4.1), propose a methodology to estimate their impact using a monolingual dataset (§4.2), and present the estimates that we obtain for various tasks and target languages (§4.3).

### 4.1 Sources of cross-lingual transfer gap

We next dissect the specific sources of cross-lingual transfer gap that each approach is sensitive to:

**Translate-test.** All the degradation comes from MT, as no multilingual model is used. We distinguish between **(i) the information lost** in the translation process (either caused by translation errors or superficial patterns removed by MT), and **(ii) the**

**distribution shift** between the original data seen during training and the machine translated data seen during evaluation (e.g., stylistic differences like *translationese* that existing models might struggle generalizing to, even if no information is lost).

**Zero-shot.** All the degradation comes from the multilingual model, as no MT is used. We distinguish between **(i) source-language representation quality**[6] relative to a monolingual model (English-only models being typically stronger than their multilingual counterparts, but only usable with *translate-test*), **(ii) target-language representation quality** relative to the source language (the representations of the target language—typically less-resourced—being worse than those of the source language), and **(iii) representation misalignment** between the source and the target language (even when a model has a certain capability in both languages, there can be a performance gap when generalizing from the source to the target language if the languages are not well-aligned).

**Translate-train.** The degradation comes from both MT and the multilingual model. However, while both source and target language representation quality have an impact,[7] this approach is not sensitive to representation misalignment, as the model is trained and tested in the same language. Regarding MT, there is no translation and therefore no information lost at test time, so we can consider potential translation errors at training time to be further inducing a distribution shift.

Finally, there can also be **an inherent distribution mismatch across languages** in the benchmark itself (e.g., the source language training data and the target language evaluation data having different properties). This can be due to annotation artifacts in multilingual datasets, in particular those

---

[6] We consider that a pretrained model A has learned higher quality representations than model B if fine-tuning it in our target task results in better downstream performance. In this case, English-only models like RoBERTa are generally superior to their multilingual counterparts like XLM-R when evaluated on downstream English tasks, so we say that their source-language representation quality is higher.

[7] Even if the model is trained and tested on the target language, we consider that there is still a degradation from the source-language representation quality under our framework. This is because we are measuring the degradation from using a multilingual model in the target language as opposed to a monolingual model in the source language, which is decomposed into the difference between the monolingual and multilingual model in the source language, plus the difference between the source and the target language in the multilingual model.

created through translation (Artetxe et al., 2020), but can also be a result of the task having naturally different properties in different languages (e.g., question answering datasets in different languages might cover different topics that are relevant to their corresponding speaker communities). Quantifying this factor would require comparing translated and original annotations for each task, which we do not consider in our analysis given the lack of such data.

## 4.2 Methodology

Let $T_{src}$ be some training data in the source language, $T_{tgt} = MT_{src \to tgt}(T_{src})$ its MT-generated translation into the target language, and $T_{bt} = MT_{tgt \to src}(T_{tgt})$ its MT-generated translation back into the source language. Similarly, let $E_{src}$, $E_{tgt}$ and $E_{bt}$ be analogously defined evaluation sets. We define $\mathrm{acc}(T, E)$ as the accuracy of our multilingual model when training on $T$ and evaluating on $E$, and $\mathrm{acc}_{mono}(T, E)$ as the accuracy of its equivalent monolingual model when training on $T$ and evaluating on $E$. Given this, we estimate the cross-lingual gap from each of the sources discussed in §4.1 as follows:

**MT information lost.** We take the difference between training and testing on original data, and training and testing on back-translated data. Given that there is not a distribution shift between train and test induced by MT, the difference in performance can be solely attributed to the information lost. MT is used twice when translating into the target language and then back into the source language, so we make the assumption that each of them introduces a similar error and divide the total gap by two to estimate the impact of a single step:

$$\Delta_{info}^{MT} = \frac{\mathrm{acc}(T_{src}, E_{src}) - \mathrm{acc}(T_{bt}, E_{bt})}{2}$$

**MT distribution shift.** We take the difference between training and testing on back-translated data, and training on original data and evaluating on back-translated data. Similar to $\Delta_{info}^{MT}$, we divide this difference by two, assuming that each MT step introduces the same error:

$$\Delta_{dist}^{MT} = \frac{\mathrm{acc}(T_{bt}, E_{bt}) - \mathrm{acc}(T_{src}, E_{bt})}{2}$$

**Source representation quality.** We take the difference between the monolingual and the multilingual model, training and testing on original data:

$$\Delta_{src}^{rep} = \mathrm{acc}_{mono}(T_{src}, E_{src}) - \mathrm{acc}(T_{src}, E_{src})$$

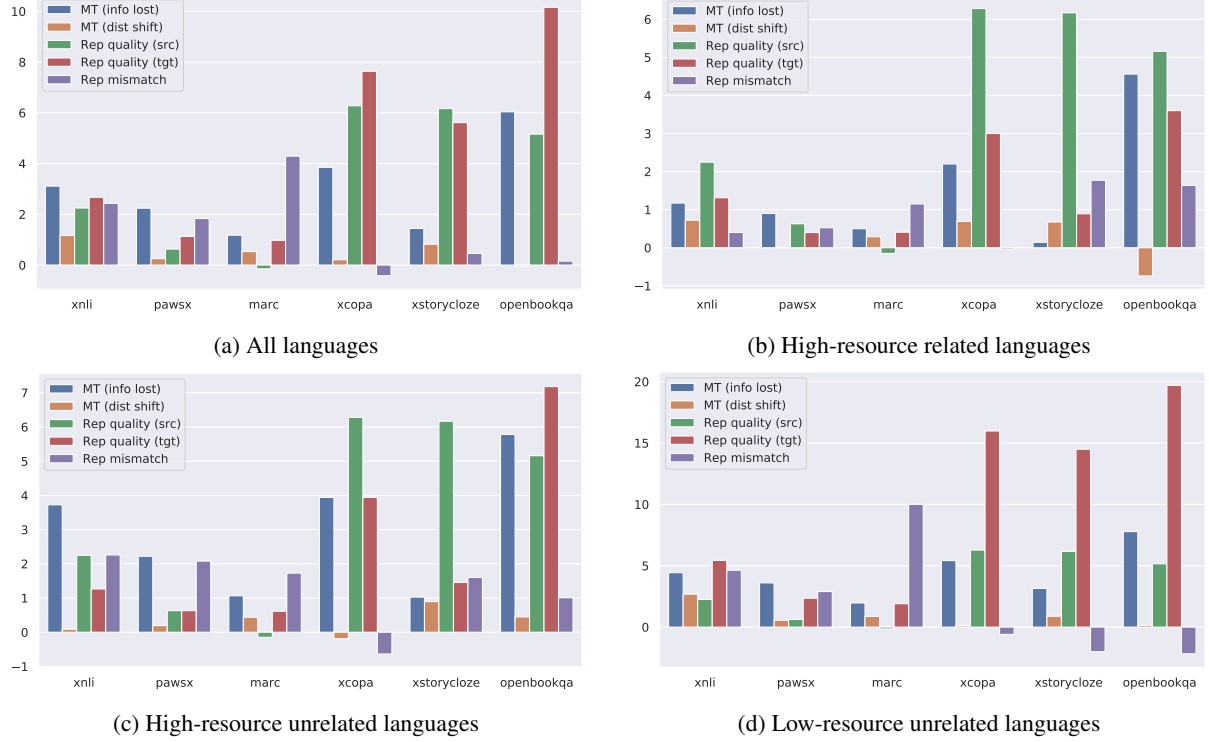

(a) All languages

(b) High-resource related languages

(c) High-resource unrelated languages

(d) Low-resource unrelated languages

Figure 2: Estimation of the impact of different sources of cross-lingual transfer gap.

**Target representation quality.** We take the difference between training and testing on original data, and training and testing on translated data, and further subtract $\Delta_{info}^{MT}$ to account for the error from MT:

$$\Delta_{tgt}^{rep} = \mathrm{acc}(T_{src}, E_{src}) - \mathrm{acc}(T_{tgt}, E_{tgt}) - \Delta_{info}^{MT}$$

**Representation misalignment.** We take the difference between training and testing on translated data, and training on original data and testing on translated data, and further subtract $\Delta_{dist}^{MT}$ to account for the error from the distribution mismatch in the later:

$$\Delta_{align}^{rep} = \mathrm{acc}(T_{tgt}, E_{tgt}) - \mathrm{acc}(T_{src}, E_{tgt}) - \Delta_{dist}^{MT}$$

### 4.3 Results and discussion

We experiment on our usual set of 6 tasks, using the exact same training data as in §3 as $T_{src}$, and the English portion of the respective test sets as $E_{src}$. EXAMs does not have an English test set, so we use the analogous OpenBookQA (Mihaylov et al., 2018) instead. We use XLM-R as our main model and RoBERTa as our monolingual model. We explore a diverse set of target languages: 5 high-resource languages related to

English (German, Dutch, French, Spanish, Italian), 5 high-resource unrelated languages (Turkish, Vietnamese, Japanese, Finnish, Arabic), and 5 low-resource unrelated languages (Malagasy, Oromo, Javanese, Uyghur, Odia). Figure 2 reports the estimates that we obtain,[8] and we next summarize our main findings:

**Degradation from MT.** We find that the impact of MT greatly varies across tasks. After manual inspection, we believe that the most relevant factors causing these differences are the format, linguistic complexity and domain of each task. For instance, degradation from MT is most prominent in OpenBookQA, a dataset of multiple choice science questions. We find two reasons for this: (i) the questions and answers often contain domain-specific terminology that is hard to translate, and (ii) the answer candidates are often unnatural or lack the necessary context to be translated in isolation. In contrast, each example in XStoryCloze consists of 5 short and simple sentences that form a story, and we find the impact of MT to be small in this dataset. As expected, MT has a larger abso-

---

[8]Different from our previous experiments, we do not perform multiple finetuning runs with different random seeds due to the high computational cost. However, our results are averaged across languages—each of them requiring separate finetuning runs.

lute impact in distant and low-resource languages. While degradation from MT comes predominantly from information lost during the translation process, a non-negligible part can be attributed to the distribution shift, which we addressed in §3.1.

**Degradation from the multilingual model.** The impact of both the source and target representation quality greatly varies across tasks, and the two are highly correlated. This suggests that there are certain tasks for which representation quality (both source and target) is generally important. By definition, the degradation from the source representation quality is invariant across languages, but the degradation from the target representation quality becomes dramatically higher for unrelated and, more importantly, low-resource languages. In general, we find that tasks requiring commonsense or factual knowledge, like XStoryCloze and Open-BookQA, are heavily sensitive to source and target representation quality, whereas shallower tasks like sentiment analysis (MARC) are barely affected. Finally, we find that, in most tasks, representation misalignment has a relatively small impact compared to representation quality. This suggests that multilingual models learn well-aligned representations that allow cross-lingual transfer at fine-tuning time, but puts into question the extent to which cross-lingual transfer is happening at pre-training time, as the target language representations can be considerably worse than the source language representations.

**Explaining the variance across tasks.** Our results can explain why the optimal cross-lingual learning approach is highly task dependent as observed in §3.3. For instance, we find that source representation quality is the most important factor in XStoryCloze, but does not have any impact in MARC. This explains why *translate-test*—the only approach that is not sensitive to this source of transfer gap—obtains the best results on XStoryCloze, but lags behind other approaches on MARC. Similarly, we find that the impact of the MT distribution shift is highest on XNLI, which is also the tasks at which our improved *translate-test* approach mitigating this issue brings the largest improvements. We remark that our analysis is only using the English portion of the benchmarks. This shows that it is feasible to characterize the cross-lingual learning behavior of downstream tasks even if no multilingual data is available.

# 5 Related work

While using MT for cross-lingual classification is a long standing idea (Fortuna and Shawe-Taylor, 2005; Banea et al., 2008; Shi et al., 2010; Duh et al., 2011), there is relatively little work focusing on it in the era of language model pretraining. Ponti et al. (2021) improve *translate-test* by treating the translations as a latent variable, which allows them to finetune the MT model for the end task through minimum risk training and combine multiple translations at inference. Our approach is simpler, but obtains substantially better results on XNLI and PAWS-X. Artetxe et al. (2020) explore a simpler variant of our training data adaptation (§3.1.2), but their focus is on translation artifacts and our numbers are considerably stronger. Oh et al. (2022) show that *translate-train* and *translate-test* are complementary and better results can be obtained by combining them. Isbister et al. (2021) report that *translate-test* outperforms monolingual models fine-tuned on the target language, but their work is limited to sentiment analysis in Scandinavian languages. While we focus on classification tasks, there is also a considerable body of work exploring MT for cross-lingual sequence labeling, which has the additional challenge of projecting the labels (Jain et al., 2019; Fei et al., 2020; García-Ferrero et al., 2022a,b)

# 6 Conclusions

Contrary to the conventional wisdom in the area, we have shown that *translate-test* can outperform both *zero-shot* and *translate-train* in most classification tasks. While most research in cross-lingual learning pivots around multilingual models, these results evidence that using an English-only model through MT is a strong—and often superior—alternative that should not be overlooked. However, there is no one method that is optimal across the board, as not all tasks are equally sensitive to the different sources of cross-lingual transfer gap. Using a new approach to quantify such sources of transfer gap, we find evidence that complex tasks like commonsense reasoning are more sensitive to representation quality, making them more suitable for *translate-test*, whereas shallower tasks like sentiment analysis work better with multilingual models. In the future, we would like to extend our study to other types of NLP problems like generation and sequence labelling, and study how the different approaches work at scale.

## Limitations

Our study is limited to classification tasks, an important yet incomplete subset of NLP problems. *Translate-train* and *translate-test* can also be applied to other types of problems like generation or sequence labeling, but require additional steps (e.g. projecting labels in the case of sequence labeling). We believe that it would be interesting to conduct similar studies on these other types of problems.

At the same time, most multilingual benchmarks—including some used in our study—have been created through translation, which prior work has shown to suffer from annotation artifacts (Artetxe et al., 2020). It should be noted that the artifacts characterized by Artetxe et al. (2020) favor *translate-train* and harm *translate-test*, so we believe that our strong results with the latter are not a result of exploiting such artifacts. However, other types of more subtle interactions might be possible (e.g. translating a text that is itself a translation might be easier than translating an original text). As such, we encourage future research to revisit the topic as better multilingual benchmarks become available.

Finally, our experiments are limited to the traditional pretrain-finetune paradigm with encoder-only models. There is some early evidence that *translate-test* also outperforms current multilingual autoregressive language models (Lin et al., 2021; Shi et al., 2022), but further research is necessary to draw more definitive conclusions.

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
