# OpenReview forum: "Revisiting Machine Translation for Cross-lingual Classification"
_EMNLP/2023/Conference — EMNLP 2023 Main_

### Official Review · Reviewer_chth · 2023-08-01

**Soundness:** 4

**Excitement:**

4: Strong: This paper deepens the understanding of some phenomenon or lowers the barriers to an existing research direction.

**Paper Topic And Main Contributions:**

The paper assesses the role of MT in cross-lingual classification by comparing the two most prominent ways of using it: translating the training data into the target language and finetuning the classification model on machine-translated target language data (“translate-train”), or translating the test set into English so as to use a classification model finetuned on English data (“translate-test”).

By means of systematic experiments, the paper shows, unlike previous works in the literature, that translate-test can outperform translate-train in many tasks provided that the output of the MT system is adapted to be similar to the English data used for finetuning, or the data used for finetuning resembles the output of the MT system.

The paper also tries to estimate where the loss of performance in classification comes from when moving to a language different from English, defining 5 categories and computing their influence by simple subtraction operations between model accuracies, obtaining different relative impacts of the categories for different tasks.

The main contributions are the following:
- Re-assessment of the different strategies for using MT for cross-lingual classification
- Definition of simple but effective methods for adapting MT for cross-lingual classification
- Outlining the type of classification tasks that more suitable for each method (translate-train and translated-test)

**Questions For The Authors:**

Question A: Could you explain the meaning of the negative values in Figure 2 for MT distribution shift and Representation misalignment?
Question B: Could you explain the motivation behind the formulas depicted for MT distribution shift, Target representation quality and Representation misalignment?

**Reasons To Accept:**

The paper contains very useful advice for building text classification systems for low-resource languages.

**Reasons To Reject:**

The analysis of the variance across tasks (section 4) lacks theoretical motivation. The analysis leverages the concept of cross-lingual transfer gap, i.e., comparing the classification performance between English and the target language, and performs an oversimplification by assuming that the impact of the different sources of degradation can be directly summed up. Unless it is empirically proved in some way, it is safe to think that interactions between the different factors that degrade quality during cross-lingual transfer is not so simple. Some of the sources of degradation have a negative value as depicted in Figure 2: this fact is not explained by the authors and casts doubt on whether the way of calculating the impact of each factor is appropriate.

**Reproducibility:**

5: Could easily reproduce the results.

**Reviewer Confidence:**

3: Pretty sure, but there's a chance I missed something. Although I have a good feel for this area in general, I did not carefully check the paper's details, e.g., the math, experimental design, or novelty.

---

> ### Author Rebuttal · Authors · 2023-08-28
>
> Thanks for the detailed review, we next address the questions that were asked:
>
> > The analysis of the variance across tasks (section 4) lacks theoretical motivation [...] and performs an oversimplification by assuming that the impact of the different sources of degradation can be directly summed up
>
> We agree that there are simplifying assumptions in our analysis. In fact, we intentionally use the term “estimate” throughout the entire section, as we are approximating the impact of each factor rather than computing a ground truth value. Having said that, we believe that such simplifying assumptions are unavoidable in this type of analysis. We think that the methodology we propose is sensible and can provide interesting insights despite the (unavoidable) simplification. To give an example, we do not claim that the degradation from MT can be exactly computed using the formulas we propose, but our analysis does provide strong evidence that some tasks like OpenBookQA are much more sensitive to this type of degradation than others like XStoryCloze. We will clarify this point in the final version of the paper.
>
> > Could you explain the meaning of the negative values in Figure 2 for MT distribution shift and Representation misalignment?
>
> The values we report should be taken as an estimate or approximation rather than an exact ground truth value for various reasons. First of all, our framework makes some simplifying assumptions as rightfully pointed out by the reviewer and discussed in the previous point. In addition, we use a finite sample to estimate these values, so there is some variance. While it is true that we obtain a few negative values, they are all small, and we think that they can be attributed to such noise.
>
> > Could you explain the motivation behind the formulas depicted for MT distribution shift, Target representation quality and Representation misalignment?
>
> The MT distribution shift accounts for the mismatch between the train and test distribution induced by MT. For that reason, we take the difference in the back-translated evaluation accuracy between training on original data (so there is a distribution shift between train and test induced by MT) and training on back-translated data (so there is no distribution shift between train and test induced by MT). MT is used twice in back-translation, so we divide the resulting value by two to estimate the impact of a single step.
>
> For the target representation quality, we take the difference between training and testing in the source language and training and testing in the target language. Say we obtain an accuracy of 80% in English and 70% in Swahili. The resulting difference of 10% can be attributed to the Swahili representations being worse than the English ones, which is what we refer to as the target representation quality. However, given that the target language data is generated through MT, which can introduce some errors due to the information lost in the translation process, we further subtract this value as estimated separately.
>
> Regarding the representation misalignment, we take the difference in the target language accuracy between training on the target language itself and training on the source language. Say we obtain an accuracy of 70% in Swahili when the training data is also in Swahili, and an accuracy of 65% when the training data is in English. The resulting difference of 5% can be attributed to the misalignment between English and Swahili in the multilingual model or, in other words, the degradation from generalizing from English to Swahili. However, given that the target language data is generated through MT, but the source language training data is not, we further need to subtract the value from the MT distribution shift as estimated separately.

---

### Official Review · Reviewer_S3YL · 2023-08-04

**Soundness:** 5

**Excitement:**

4: Strong: This paper deepens the understanding of some phenomenon or lowers the barriers to an existing research direction.

**Missing References:**

- "Is Machine Translation Ripe for Sentiment Analysis", ACL 2011, http://www.aclweb.org/anthology/P11-2075 - this is an older paper that focuses on cross-lingual sentiment classification, but it also investigates both MT and domain issues like you did in this paper. It may be interesting to refer to.
- There have been much discussions on "translate-test" vs "translate-train" in the context of Cross-Language Information Retrieval. It is a very different problem than NLI or classification, but just for your reference. See for example the book by Nie: https://link.springer.com/book/10.1007/978-3-031-02138-1



**Paper Topic And Main Contributions:**

This paper re-examines cross-lingual classification for the XNLI task by using stronger MT systems. In particular, it revisits the discussion about the relative merits of the translate-test, translate-train, and zero-shot approaches. One important conclusion, in my opinion, is that the translate-test approach is more competitive than previously thought; while I'm not sure if this was the common sentiment in the field, in any case it suggests that stronger MT could be helpful and the multilingual pretraining of XNLI models may be less important. This suggestion by itself is a thought-provoking idea and I think it benefits the field to have "revisiting" papers like this.

Generally, I think the paper is written very clearly. The approach is very methodological, and each experimental design decision is justified. The main contributions are (1) the stronger MT engines, including the doc-level and roundtrip adaptation, (2) the main results in Table 4, and (3) the analysis in Section 4.

I did not get so much out of the results of Section 4 (i.e. Fig 2 is challenging to interpret and some might draw different conclusions from yours), though I do very much like the way the authors approach different kinds of degradation (Sec 4.2). I would give this to new PhD students as a good reference for how to think about error analysis.

**Questions For The Authors:**

- For the MT engines, do you train a single adapted MT per dataset (e.g. xnli) or together for all six datasets

**Reasons To Accept:**

- Thought-provoking claims on the relative strengths of train-test and importance of multilingual pretraining.
- Nice small contribution to better MT for XNLI like tasks
- Well-designed methodology and clear writing that makes this "revisitation" paper an interesting read
- Exemplar analysis framework

**Reasons To Reject:**

None

**Reproducibility:**

4: Could mostly reproduce the results, but there may be some variation because of sample variance or minor variations in their interpretation of the protocol or method.

**Reviewer Confidence:**

4: Quite sure. I tried to check the important points carefully. It's unlikely, though conceivable, that I missed something that should affect my ratings.

**Typos Grammar Style And Presentation Improvements:**

- I think a figure on the different ways you generated data for MT would be helpful. It is relatively clear from the text (Sec 3.1.1 and 3.1.2), but it took me a few readings. Specifically, a figure indicating examples of how sentences are backtranslated, then forward-translated, and which parts are used for which adaptation system would be helpful.
- I'm not sure if "doc-level" is a good term because in MT it refers to a real document involving many sentences. How about pair-level or something that emphasizes the NLI aspect?

---

> ### Author Rebuttal · Authors · 2023-08-28
>
> Thanks for the insightful and encouraging review. We address the main points raised next:
>
> > For the MT engines, do you train a single adapted MT per dataset (e.g. xnli) or together for all six datasets
>
> We train a separate adapted MT per dataset.
>
> > Missing references
>
> Thanks for the suggested references. Our current version of the related work section focuses on recent work that is more directly relevant to our work, but we will extend it to give a broader perspective, covering the papers suggested by the reviewer.
>
> > I think a figure on the different ways you generated data for MT would be helpful
>
> Thanks for the suggestion, which we will consider for the camera ready version.
>
> > I'm not sure if "doc-level" is a good term because in MT it refers to a real document involving many sentences. How about pair-level or something that emphasizes the NLI aspect?
>
> We agree that, in the MT literature, “doc-level” usually refers to a real document involving many sentences. In our case, we have various fields in some of the tasks (e.g., the premise and hypothesis in NLI), but in some instances each of these fields can contain more than one sentence too. For instance, MARC consists of Amazon reviews, which typically contain many sentences. For the baseline system, we run sentence segmentation and translate each of these sentences separately, whereas for the doc-level variant we translate the entire input as a whole, which can involve both more than one field and more than one sentence. For that reason, we believe that “pair-level” would not be an appropriate term. Having said that, we understand that this is not entirely clear in the current version of the paper, and we will clarify it for the camera ready version.

---

### Official Review · Reviewer_p9fP · 2023-08-10

**Soundness:** 4

**Excitement:**

4: Strong: This paper deepens the understanding of some phenomenon or lowers the barriers to an existing research direction.

**Paper Topic And Main Contributions:**

This paper focuses on cross-lingual classification, and demonstrates how machine translation approaches can prove very effective. In particular, the authors argue that translating the test set into English and using English monolingual PLMs can constitute a much stronger baseline than previously thought. Lastly, the authors also introduce a methodology for studying cross-lingual transfer gaps.

**Reasons To Accept:**

- The paper is very cogent and well written,. The results they provide clearly exhibit that a more careful formulation of MT-based approaches to cross-lingual classification can bring about significant improvements over more naive approaches
- The methodology introduced in §4.2 to evaluate which factors contribute to cross-lingual transfer gaps can be of interest to the community in general

**Reasons To Reject:**

**A/** There are some methodological flaws in the current state of the paper, in particular:
- ~As far as I can tell, the authors only report experiments across 1 seed~ I was mistaken and appreciate the authors' pointing this out.
- Some of the margins of improvements in quantitative experiments (tables 1, 2, 3 and 4) are slim enough that they call for significance testing

On a related though less important note:
- Footnote 5 suggest that one specific experiment (§3.1.2) corresponds to a much larger training set than other baselines, making it more difficult to compare results across experiments
- In the definition of $\Delta^{MT}_{info}$  the authors assume that test and training data will have comparable distributions.
  + This overlooks potential train test mismatches or biases that could in principle be mitigated or obfuscated by the backtranslation procedure
  + This is problematic, as the same value is then used to calculate $\Delta^{rep}_{tgt}$.

**B/** On a more theoretical level, I'm having some difficulties in conceptualizing what is the relevance of a translate-test approach.
- If one considers cross-lingual datasets like XNLI as tools for benchmarking the cross-lingual capabilities of a given model, then a translate-test approach, by converting the task at hand to a monolingual one, ends up being at most marginally relevant to cross-lingual studies. It may constitute a supplementary baseline but I fail to see how it could advance the field's understanding of how to develop efficient multilingual representations.
- Perhaps a more fruitful line of thought would be to consider cross-lingual datasets like XNLI as engineering problem &mdash; some toy example of a real-world situation where large quantities of data are available in one language (English), but not in the lower-resource target language of interest. Under such a view, I assume that translate test might have an interest, but the paper in its current format does not present relevant point of comparisons: either the study should focus on low resource languages where no pretrained model is available, or the study should compare a translate-test baseline to target-language monolingual approaches, rather than the purely English-centric point of view.

Long story short: I am uncertain what point the authors are trying to make. It seems the paper would benefit from a more explicit stance as to what is the intended reach of the proposed approach, and how relevant it is for the intended readership.

**Reproducibility:**

4: Could mostly reproduce the results, but there may be some variation because of sample variance or minor variations in their interpretation of the protocol or method.

**Reviewer Confidence:**

3: Pretty sure, but there's a chance I missed something. Although I have a good feel for this area in general, I did not carefully check the paper's details, e.g., the math, experimental design, or novelty.

---

> ### Author Rebuttal · Authors · 2023-08-28
>
> Thanks for the detailed review. We next address the various points that were raised:
>
> > As far as I can tell, the authors only report experiments across 1 seed.
>
> This is not accurate, we actually report results across 5 seeds as described in Section 2 (L133-135: *we do 5 finetuning runs with different random seeds, and report accuracy numbers in the test set averaged across all languages and runs*).
>
> > Some of the margins of improvements in quantitative experiments are slim
>
> We agree that the differences in some datasets are small, but we do not think this contradicts our claims, which we believe are well supported. In fact, one of our main conclusions is that *there is a considerable variance across tasks* (L360-361), and *the optimal cross-lingual learning approach is highly task dependent* (L371-372), which also includes cases where two approaches perform similarly.
>
> > one specific experiment (§3.1.2) corresponds to a much larger training set than other baselines, making it more difficult to compare results across experiments
>
> While it is true that this variant uses a larger training set, it is still comparable in terms of the human supervision it uses. More concretely, this variant uses the exact same set of original examples as the others, and all the additional examples are automatically derived from them using MT-based paraphrasing. As such, it does not require any additional (human-labelled) data. This is similar to how translate-train works, which generates a larger training set by translating the original English examples into all other languages and combining them all. This other approach is widely used in the literature, and also covered in our experiments.
>
> > the authors assume that test and training data will have comparable distributions
>
> When defining the MT information lost, the paper says that the training and test data come from the same distribution, but what we actually meant is that there is no distribution shift between the training and test set induced by translation, which is true by design. It is not necessary to assume that the training and test data come from the exact same distribution, so we realize that our statement is misleading. Thanks for pointing it out, we will fix it in the camera ready version!
>
> > I'm having some difficulties in conceptualizing what is the relevance of a translate-test approach
>
> We think that research on translate-test can be relevant from multiple perspectives.
>
> First of all, translate-test has an inherent practical interest as an alternative to multilingual pretrained models. While we naturally focus on academic benchmarks, similar multilingual classification problems are very common in industry. Our improvements to translate-test, and the finding that this approach can be competitive and even superior to others in most cases, are thus directly relevant to solve such practical multilingual classification tasks in the real world.
>
> Besides that, while it is true that our work does not propose a new method to learn better multilingual representations, we believe that having strong baselines is important to advance the field. In addition, our analysis in Section 4 shows that different tasks have different properties that explain why existing multilingual models do better or worse in them, and understanding this can be useful to develop better multilingual models in the future.

---

### Meta-Review · Area_Chair_6G5e · 2023-09-17

**Recommendation:** 4

**Metareview:**

The paper finds that "translate-test" could be a stronger method than what people previously expected. The reviewers appreciate the comprehensive analysis on various classification tasks and the paper is well written. The paper could be further improved by provide more theoretical justification of the results. Moreover, it would be useful to clarify the languages included in the task in the paper. It might be interesting to understand whether this conclusion still holds for very under-represented languages other than those studied in the paper, or if it still holds for tasks other than classification.

---

### Decision · Program_Chairs · 2023-10-07

**Decision:**

Accept-Main

**Comment:**

The paper finds that "translate-test" could be a stronger method than what people previously expected. The reviewers appreciate the comprehensive analysis on various classification tasks and the paper is well written. The paper could be further improved by provide more theoretical justification of the results. Moreover, it would be useful to clarify the languages included in the task in the paper. It might be interesting to understand whether this conclusion still holds for very under-represented languages other than those studied in the paper, or if it still holds for tasks other than classification.